# Back flux during anaerobic oxidation of butane supports archaea-mediated alkanogenesis

Song-Can Chen [1,2], Sheng Chen[3], Niculina Musat [4], Steffen Kümmel [5], Jiaheng Ji[5], Marie Braad Lund[4], Alexis Gilbert [6], Oliver J. Lechtenfeld [7], Hans-Hermann Richnow [5] & Florin Musat [4,8] ✉

Microbial formation and oxidation of volatile alkanes in anoxic environments significantly impacts biogeochemical cycles on Earth. The discovery of archaea oxidizing volatile alkanes via deeply branching methyl-coenzyme M reductase variants, dubbed alkyl-CoM reductases (ACR), prompted the hypothesis of archaea-catalysed alkane formation in nature (alkanogenesis). A combination of metabolic modelling, anaerobic physiology assays, and isotope labeling of *Candidatus* Syntrophoarchaeum archaea catalyzing the anaerobic oxidation of butane (AOB) show a back flux of $CO_2$ to butane, demonstrating reversibility of the entire AOB pathway. Back fluxes correlate with thermodynamics and kinetics of the archaeal catabolic system. AOB reversibility supports a biological formation of butane, and generally of higher volatile alkanes, helping to explain the presence of isotopically light alkanes and deeply branching ACR genes in sedimentary basins isolated from gas reservoirs.

Biological formation and oxidation of volatile alkanes, particularly methane, have a major impact on biogeochemical cycles on Earth. Carried out by methanogenic archaea, methanogenesis is one of the major processes of biomass degradation in deep, anoxic sediment horizons, generating vast amounts of methane, a potent greenhouse gas[1]. Most of the formed methane, along with methane released from thermogenic reservoirs, is oxidized already in anoxic sediments. The anaerobic oxidation of methane (AOM) is carried out by anaerobic methanotrophic archaea (ANME) essentially via the reverse methanogenesis pathway[2]. The key enzyme of both pathways is methyl-coenzyme M reductase (MCR), which catalyzes both the release of methane in methanogenesis, and the initial step of methane oxidation in AOM[3]. The pathways of methanogenesis and AOM operate close to thermodynamic equilibrium, and are fully reversible, with

demonstrated back fluxes of reaction products into the substrate pools[2,4,5]. Recent findings showed that ANME-related archaea can oxidize higher, $C_{2+}$ alkanes using biochemical mechanisms similar to those driving the AOM[6–10]. The anaerobic oxidation of $C_{2+}$ alkanes (AOAlk) is apparently initiated by deeply branching MCR variants, dubbed alkyl-coenzyme M reductases (ACR)[6,8,11]. ACRs are structurally similar to MCR, and similar enzymatic mechanisms are expected. Notable differences include amino acids substitutions leading to wider catalytic chambers suited to accommodate substrates bulkier than methane[3,12,13]. A potential reversibility of AOAlk, suggested by enzyme similarities and shared pathway modules with AOM[6], is supported by the unspecific conversion of ethyl-CoM to ethane by MCRs of methanogens, and by back flux studies with an archaea-SRB consortium catalyzing the anaerobic oxidation of ethane (AOE)[7,14].

[1]Division of Microbial Ecology, Center for Microbiology and Environmental Systems Science, University of Vienna, Vienna, Austria. [2]MOE Key Laboratory of Environment Remediation and Ecological Health, College of Environmental and Resource Sciences, Zhejiang University, Hangzhou, China. [3]Research Center for Mathematics, Advanced Institute of Natural Sciences, Beijing Normal University, Zhuhai, China. [4]Department of Biology, Section for Microbiology, Aarhus University, Aarhus, Denmark. [5]Department of Technical Biogeochemistry, Helmholtz Centre for Environmental Research – UFZ, Leipzig, Germany. [6]Department of Earth and Planetary Sciences, Tokyo Institute of Technology, Meguro, Tokyo, Japan. [7]Department of Environmental Analytical Chemistry, Helmholtz Centre for Environmental Research – UFZ, Leipzig, Germany. [8]Department of Molecular Biology and Biotechnology, Faculty of Biology and Geology, Babeș-Bolyai University, Cluj-Napoca, Romania. ✉e-mail: florin.musat@bio.au.dk

Here we show that the entire archaeal pathway for the anaerobic oxidation of butane (AOB) is reversible by measuring a steady back flux of carbon from the $CO_2$ to the butane pools during net AOB. Experiments were done with a thermophilic AOB enrichment culture, culture Butane50, obtained from Guaymas Basin hydrothermal vent sediments[8]. This culture consists of two archaeal species, *Candidatus* Syntrophoarchaeum butanivorans and *Ca*. S. caldarius, which form tightly-packed aggregates with partner sulfate-reducing bacteria (SRB)[8]. Oxidation of butane was assumed to be carried out by the more abundant *Ca*. S. butanivorans, while the role of *Ca*. S. caldarius was unclear. The partner SRB were apparently scavenging reducing equivalents generated during butane oxidation, which are used to reduce sulfate to sulfide. The net reaction resembles AOB by anaerobic bacteria which couple butane oxidation to $CO_2$ and sulfate reduction to sulfide in the same cell, with similar net stoichiometries and energy yields[15,16].

## Results

### *Candidatus* Syntrophoarchaeum butanivorans as the dominant butane oxidizer

The Butane50 culture was continuously grown in mineral medium with butane as the sole growth substrate. During consecutive transfers in fresh culture media, the culture grew predominantly as archaea-bacteria aggregates (Fig. 1A), as formerly described[8]. Amplicon sequencing showed a reduced diversity (Supplementary Fig. 1), with only 10 lineages with relative abundance >1%, typical of enrichment cultures grown under defined conditions[16,17]. *Ca*. Desulfofervidus and *Ca*. Syntrophoarchaeum were recovered at 30% and 11% relative abundance, respectively (Supplementary Fig. 1). A role of bacteria in oxidation of butane was excluded based on the lack of signature metabolites (butylsuccinates) and of genes typical for a bacterial AOB, or generally of alkanes (methylalkylsuccinate or alkylsuccinate synthases; *mas/ass*)[8,18]. Repeated metabolomics analyses showed the presence of butyl-coenzyme M in cellular metabolite pools, confirming that AOB in the Butane50 culture is carried out by archaea[8]. To resolve the role of *Ca*. S. butanivorans and *Ca*. S. caldarius in butane oxidation, the abundance of the two strains in the Butane50 culture was determined by hybridization with species-specific oligonucleotide probes. Direct cell counts showed that *Ca*. S. butanivorans accounted on average for about 50% of the total cell numbers, and over 95% of all archaea (Fig. 1B, C and Supplementary Fig. 2). *Ca*. S. caldarius represented, on average, about 2% of the total cell number, and about 5% of all archaea. Due to its low cell abundance and presence of *acr* genes *Ca*. S. caldarius is likely able to oxidize butane at much lower rates than *Ca*. S. butanivorans. Double hybridizations with species-specific probes

and a general archaea probe (Supplementary Table 1) showed that *Ca*. S. butanivorans and *Ca*. S. caldarius represent over 99% of the total archaea cells, excluding a role of other archaea in AOB. We conclude that *Ca*. S. butanivorans is the main butane oxidizer in the Butane50 culture, and we chose its AOB pathway for further metabolic modelling.

### Metabolic model of AOB and reverse AOB

To assess the potential reversibility of AOB, we applied a metabolic-thermodynamic model[19] to the AOB pathway operating in both forward and reverse directions. The AOB catalyzed by *Ca*. S. butanivorans was generalized as $C_4H_{10} + 13X \rightarrow 4CO_2 + 13XH_2$, with $XH_2/X$ being a generic two-electron carrier, currently unknown (Supplementary Note 1). Hereafter, we use *AOB pathway* to refer to the butane oxidation carried out by the archaea, and *AOB process* to refer to the complete reaction, including reduction of sulfate to sulfide by the partner SRB. Genome-based metabolic reconstruction showed that the overall AOB pathway consists of at least 17 intermediate steps (Table 1 and Supplementary Table 2), which we formally grouped into four metabolic modules: butane activation, conversion of butyl-CoM to butyryl-CoA, beta-oxidation, and the reverse Wood-Ljungdahl pathway (Fig. 2A and Table 1). The conversion of butyl-CoM to butyryl-CoA, which could proceed through several consecutive reactions is carried out by enzymes currently unknown (Supplementary Note 2 and Supplementary Data 1–4). Among the 26 electrons released from butane oxidation, four reach X in the conversion of butyl-CoM to butyryl-CoA, and 22 were assumed to enter the membrane-bound menaquinone (MQ) pool before delivery to X. The latter occurs via three different membrane-bound enzymatic complexes, including NADH dehydrogenase (NDH), Fqo, and Nuo, which oxidize 5 NADH, 4 $F_{420}H_2$, and 4 reduced ferredoxins ($Fd_{red}$), respectively (Fig. 2B).

The model was applied by setting the redox potential of the $XH_2/X$ redox couple to the average redox potential of sulfate reduction ($E^{0\prime} = -0.220$ V) as an upper redox potential limit, reflecting the proposed nanowire-based extracellular electron transfer mechanism between alkane-oxidizing archaea and SRB[8,20,21]. In the oxidative AOB pathway, energy conservation through the generation of proton motive force (PMF) has been proposed to take place during oxidation of $F_{420}H_2$ and $Fd_{red}$[8]. When these were fed to the model as potential energy recovery sites, the model predicted unfeasible AOB kinetics, due to low quasi-equilibrium metabolite concentrations (<1 μM) (Supplementary Fig. 3). This is likely due to energy barriers imposed by endergonic reactions, including butyl-CoM/butyryl-CoA conversion and electron transport between X and the MQ pool (Table 1). To

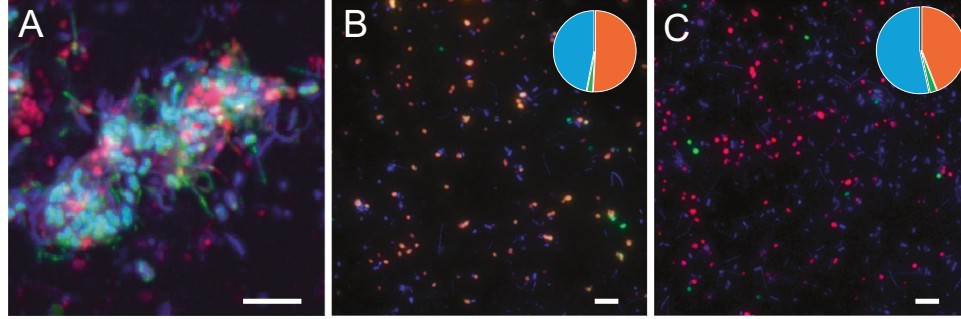

**Fig. 1 | Microscopy characterization of the Butane50 culture. A** Native aggregate of *Ca*. Syntrophoarchaeum butanivorans, *Ca*. S. caldarius (red-purple signal) and *Ca*. Desulfofervidus auxilii (blue-green signal) after double hybridization with general probes for Archaea and Bacteria and DAPI counterstaining. The Archaea appear magenta, and the Bacteria appear cyan due to overlay of probe and DAPI signals. Representative of *n* = 3 images. **B** Double hybridization of dispersed aggregates with probes specific for Archaea (green signal), and for *Ca*. S. butanivorans (orange signal; *Ca*. S. butanivorans cells were hybridized with both probes).

**C** Double hybridization of dispersed aggregates with probes specific for *Ca*. S. butanivorans (red-purple signal) and for *Ca*. S. caldarius (green signal). Each of the (**B**, **C**) are representative of *n* > 60 images from *n* = 3 independent cultures. Insets in (**B**, **C**) show the average abundance (*n* > 5000 cells counted) of *Ca*. S. butanivorans (orange), other Archaea (green, **B**), *Ca*. S. caldarius (green, **C**), and Bacteria (blue, both panels). Scale bars = 5 μm, applicable to all panels. Details of the oligonucleotide probes used are summarized in Supplementary Table 1.

**Table 1 | Changes of free energy ($\Delta G^{o'}$; pH = 7, $T$ = 298 K) for each AOB metabolic reaction**

| Index | Reaction[a] | $\Delta G^{o'}$ (kJ mol$^{-1}$) | Stoi.[c] | Enzyme[d] | $Ex$ = −0.414V[e] | $Ex$ = −0.22V[f] |
|---|---|---|---|---|---|---|
| 1 | $C_4H_{10}$ (g) + CoM-S-S-CoB == $C_4H_9$-S-CoM + HS-CoB | 16.900 | 1 | ACR | 16.900 | 16.900 |
| 2 | HS-CoM + HS-CoB + 2Fd$_{red}$ + 2H$^+$ + 2F$_{420}$ == CoM-S-S-CoB + 2Fd$_{ox}$ + 2F$_{420}$H$_2$ | 7.141 | 1 | Hdr-FrhB | 7.141 | 7.141 |
| 3 | $C_4H_9$-S-CoM + CoA + H$_2$O + 2X == Butyryl-CoA + HS-CoM + 2XH$_2$ | Related to $Ex$[b] | 1 | ? | 104.003 | 29.119 |
| 4 | Butyryl-CoA + 2NAD$^+$ + 2Fd$_{red}$ == Crotonyl-CoA + 2NADH + 2Fd$_{ox}$ | 25.090 | 1 | Bcd-Etf | 25.090 | 25.090 |
| 5 | Crotonyl-CoA + H$_2$O == (S)-3-Hydroxybutyryl-CoA | 0.300 | 1 | Ech | 0.300 | 0.300 |
| 6 | (S)-3-Hydroxybutyryl-CoA + NAD$^+$ == Acetoacetyl-CoA + NADH + H$^+$ | 12.545 | 1 | HADH | 12.545 | 12.545 |
| 7 | Acetoacetyl-CoA + CoA == 2 Acetyl-CoA | −28.100 | 1 | ACAT | −28.100 | −28.100 |
| 8 | NADH + H$^+$ + MQ == NAD$^+$ + MQH$_2$ | −46.320 | 5 | NDH | −46.320 | −46.320 |
| 9 | Acetyl-CoA + H$_4$MPT + 2 Fd$_{ox}$ + H$_2$O == CH$_3$-H$_4$MPT + CoA + CO$_2$ (g) + 2 Fd$_{red}$ + 2H$^+$ | 66.114 | 2 | ACS-CODH | 66.114 | 66.114 |
| 10 | CH$_3$-H$_4$MPT + NAD$^+$ == CH$_2$ = H$_4$MPT + NADH + H$^+$ | 1.140 | 2 | Met | 1.140 | 1.140 |
| 11 | CH$_2$ = H$_4$MPT + F$_{420}$ + H$^+$ == CH ≡ H$_4$MPT$^+$ + F$_{420}$H$_2$ | −6.500[g] | 2 | Mtd | −6.500 | −6.500 |
| 12 | CH ≡ H$_4$MPT$^+$ + H$_2$O == HCO-H$_4$MPT + H$^+$ | 2.000[g] | 2 | Mch | 2.000 | 2.000 |
| 13 | HCO-H$_4$MPT + MF == HCO-MF + H$_4$MPT | −21.100[g] | 2 | Ftr | −21.100 | −21.100 |
| 14 | HCO-MF + H$_2$O + 2Fd$_{ox}$ == CO$_2$ (g) + MF + 2H$^+$ + 2Fd$_{red}$ | 0.598 | 2 | Fwd | 0.598 | 0.598 |
| 15 | F$_{420}$H$_2$ + MQ == F$_{420}$ + MQH$_2$ | −50.180 | 4 | Fqo | −50.180 | −50.180 |
| 16 | 2 Fd$_{red}$ + 2H$^+$ + MQ == 2 Fd$_{ox}$ + MQH$_2$ | −81.060 | 2 | Nuo | −81.060 | −81.060 |
| 17 | MQH$_2$ + X == MQ + XH$_2$ | Related to $Ex$[b] | 11 | ? MHCs | 64.462 | 27.020 |

[a]X and XH$_2$ refer to the oxidized and reduced forms of the unknown electron carrier between butane oxidizing archaea and sulfate-reducing bacteria. Similarly, MQ and MQH$_2$ refer to the oxidized and reduced forms of menaquinone.

[b]The $\Delta G^{o'}$ of reactions 3 and 17 is −386$Ex$ – 55.801 and −193$Ex$ – 15.440 kJ mol$^{-1}$, respectively. $Ex$: redox potential of the unknown electron carrier X under biological standard condition (unit: V).

[c]Stoichiometry of each individual reaction step during AOB; the sum of all reactions is the net AOB $C_4H_{10}$ + 13X → 4CO$_2$ + 13XH$_2$.

[d]ACR alkyl-coenzyme M reductase, Hdr heterodisulfide reductase, FrhB F$_{420}$ hydrogenase subunit B, Bcd butyryl CoA dehydrogenase, EtfAB electron transfer proteins, Ech Enoyl-CoA hydratase, HADH 3-Hydroxyacyl-CoA dehydrogenase, ACAT Acetyl-CoA acetyltransferase, NDH NADH dehydrogenase (non-electrogenic), ACS acetyl-CoA synthase, CODH CO dehydrogenase, Met: methyl-methanotetrahydropterin (H$_4$MPT) dehydrogenase, Mch N$^5$,N$^{10}$-methenylH$_4$MPT cyclohydrolase, Ftr FormylMF-H$_4$MPT formyltransferase, Fwd Formylmethanofuran dehydrogenase, Fqo F$_{420}$H$_2$:quinone oxidoreductase, Nuo-like NADH:quinone oxidoreductase, MHCs multiheme-cytochrome c.

[e]Standard Gibbs free energy change for each reaction step assuming $Ex$ is equivalent to the standard redox potential of H$_2$/H$^+$.

[f]Standard Gibbs free energy change for each reaction step assuming $Ex$ is equivalent to the average redox potential of sulfate reduction.

[g] $\Delta G^{o'}$ of the reaction is gather from Blaut et al.[79]; $\Delta G^{o'}$ of the remaining reactions are calculated from standard redox potential or estimated using Equilibrator (Supplementary Data).

overcome these barriers, the model was run considering two endergonic reactions coupled with PMF consumption, analogous to energy-driven processes proposed in methanogenic archaea[1]. The energy investment, together with energy dissipation at the methylene-tetrahydromethanopterin oxidation step, allows a kinetically feasible AOB pathway (Fig. 2C). Depending on process conditions (i.e., varied ratios of XH$_2$/X), such a system could translocate four to six protons for net energy conservation (Fig. 2B).

We deemed this model feasible, and applied it to the opposite process direction, butane formation. In this case, the energy investment and energy harvest sites are switched: energy is needed to generate F$_{420}$H$_2$ and Fd$_{red}$, and energy is conserved as PMF during the transformation of butyryl-CoA to butyl-CoM and the electron flow from XH$_2$ to MQ. This system is predicted to be thermodynamically and kinetically feasible, provided energy is dissipated at an exergonic reaction (acetyl-CoA synthesis; Fig. 2D). Depending on the catalytic energy of the overall reaction, the reverse AOB pathway (rAOB) could harvest a net of three to six protons for anabolism and growth (Fig. 2B). We observed similar patterns when the model was run considering the redox potential of XH$_2$ equivalent to that of H$_2$ as a lowest redox potential limit ($E^{o'}$ = −0.414 V; Supplementary Fig. 4). This suggests that the predicted reversibility is robust against the choice of XH$_2$/X redox potential within a plausible range. Based on the van't Hoff equation (Supplementary Note 3) a higher potential reversibility of the entire AOB process is predicted at 50 °C, the optimum growth temperature of *Ca*. S. butanivorans, vs. the standard biochemical state temperature (25 °C) used for modelling.

## Carbon dioxide to butane backflux during net AOB

The potential reversibility returned by the model suggested that during net butane oxidation the dynamic pools of AOB intermediates,

including butane, CO$_2$ and X/XH$_2$, could generate a back flux of CO$_2$ to butane. To probe the transfer of carbon from the CO$_2$ pool into the butane pool, AOB cultures were rendered bicarbonate-free by cultivation in Tris-buffered mineral medium. Replacement of the bicarbonate buffer by Tris had no impact on butane oxidation rates. For back flux assays, Tris-buffered cultures were supplied with defined amounts of butane (6 mmol l$^{-1}$) and sulfate (28 mmol l$^{-1}$) as sole sources of carbon and energy. Labeling was done by adding 18 mmol l$^{-1}$ sodium bicarbonate containing $^{13}$C at concentrations ranging from natural abundance, or approx. 1.12 atom%, to up to 98 atom%. Under the experimental conditions used (pH = 7.3) the added bicarbonate was establishing equilibrium with CO$_2$ and HCO$_3^-$ as major species. The $^{13}$C labeling ratio was expected to decrease during the assays, as unlabeled CO$_2$ from butane oxidation was transferred into the labeled CO$_2$ pool (Supplementary Note 4 and 5). In these cultures, oxidation of butane was coupled with reduction of sulfate to sulfide. The average ratio between reducing equivalents generated by butane oxidation and consumed by sulfate reduction was 1.04, as expected for the net AOB stoichiometry (Supplementary Table 3). Addition of $^{13}$C-CO$_2$/HCO$_3^-$ (hereafter, dissolved inorganic carbon, DIC) had no noticeable impact on AOB kinetics, with oxidation of butane and reduction of sulfate proceeding at similar rates among replicate cultures (Fig. 3A). Oxidation of butane in the presence of natural $^{13}$C-abundance DIC led to an enrichment in $^{13}$C of the residual butane pool, with the isotopic composition, δ$^{13}$C, shifting from −25‰ to −15‰, in accordance with expected kinetic isotope effects of AOB (Fig. 3B). We calculated an isotopic fractionation factor, ε$^{13}$C, of −2.5 ± 0.2‰ (Supplementary Note 6 and Supplementary Fig. 5), comparable to ε$^{13}$C determined for butane-oxidizing anaerobic bacteria[22].

AOB in the presence of $^{13}$C-labeled DIC led to a progressive $^{13}$C-enrichment of the residual butane pool (Fig. 3B). The extent of

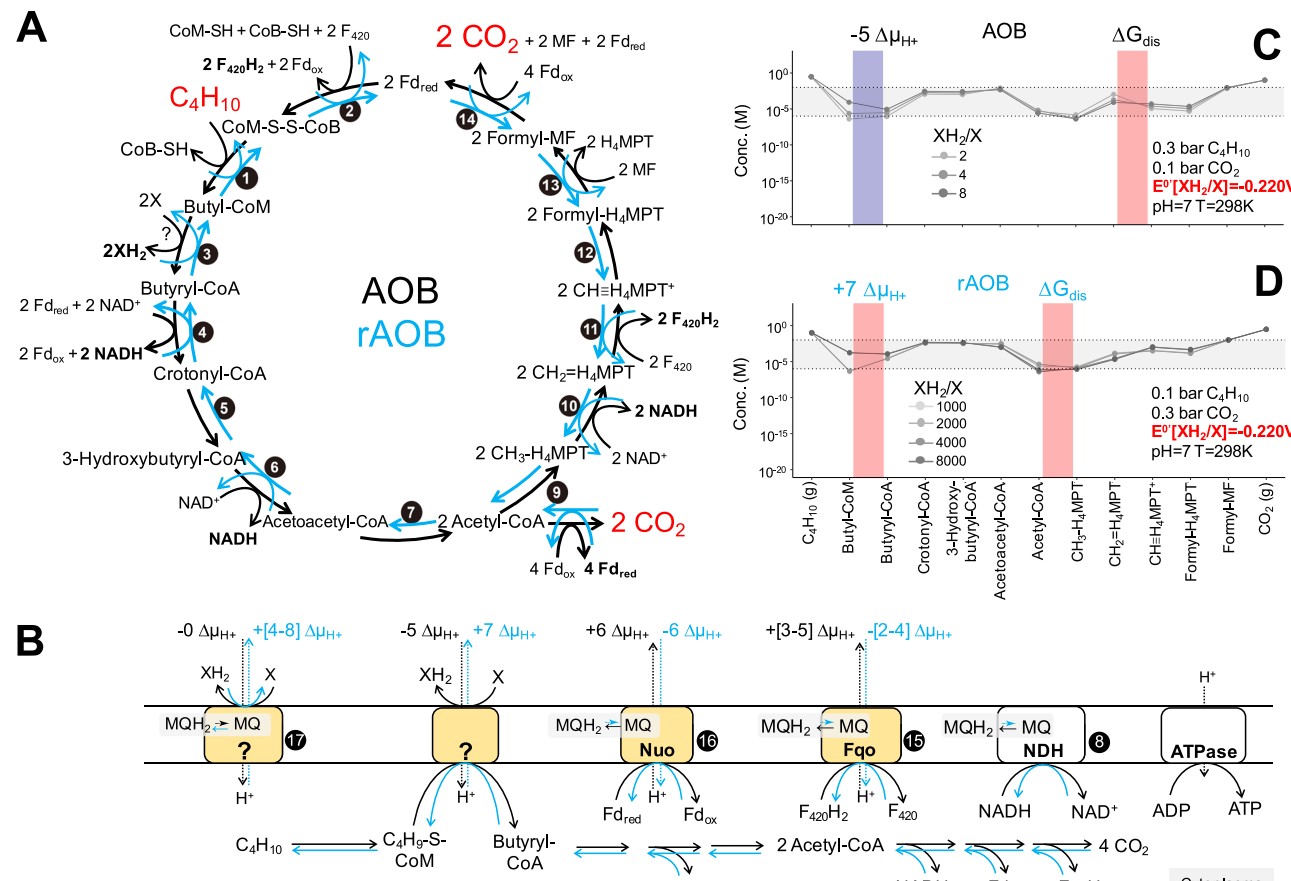

**Fig. 2 | Metabolic thermodynamic model of the anaerobic oxidation of butane (AOB) and reverse AOB.** **A** Refined central carbon and energy metabolism during AOB (black arrows) or reverse AOB (light blue arrows). Numbers in the figure correspond to reactions in Table 1. **B** The reducing equivalents released during AOB/consumed during rAOB include $2\,XH_2$, $5\,NADH$, $4\,F_{420}H_2$, and $4\,Fd_{red}$, which in total account for 26 electrons. **C**, **D** Quasi-equilibrium concentrations (mol L$^{-1}$) of AOB/rAOB metabolites considering a standard redox potential of $XH_2/X$ of $-0.220$ V. **C** The AOB model considers energy recovery at $F_{420}H_2$ and $Fd_{red}$ oxidation, energy investment at the butyl-CoM/butyryl-CoA conversion and at electron transport between MQ and X, and energy dissipation ($\Delta G_{dis}$) at oxidation of

methylene-tetrahydromethanopterin. Metabolite concentrations were calculated under $XH_2/X$ ratios of 2, 4, and 8, corresponding to AOB free energy change ($\Delta G_{cat}$) of $-147.23$, $-124.91$, and $-102.58$ kJ mol$^{-1}$ butane, respectively. **D** The rAOB model considers a switch in energy recovery and energy investment, with energy dissipation occurring at acetyl-CoA synthesis. Metabolite concentrations were calculated for $XH_2/X$ ratios of 1000, 2000, 4000, and 8000, corresponding to rAOB $\Delta G_{cat}$ of $-66.54$, $-88.87$, $-111.19$, and $-133.52$ kJ mol$^{-1}$ butane, respectively. $\Delta\mu_{H+}$ denotes the free energy required to translocate one proton across the membrane ($-20$ kJ mol$^{-1}$). Source data are provided as a Source Data file.

labeling was dependent on the initial $^{13}$C concentration in the DIC pool. The highest δ$^{13}$C-butane (approx. +80‰) was recorded in the assays with 98% initial $^{13}$C-DIC. In all assays, labeling of the residual butane pool peaked after approximately 120 days of incubation, when over 98% of the initially added butane was consumed and the oxidation of butane effectively ceased. For the rest of the incubation the δ$^{13}$C-butane and the butane concentrations remained relatively constant. Thus, the back reaction of AOB was strictly linked to the forward reaction of the process. A linear correlation between the δ$^{13}$C-butane and the amount of $^{13}$C in the DIC pool was observed for the stationary phase (Fig. 3C). Butane in sterile controls showed no δ$^{13}$C variations, excluding abiotic isotope exchanges between butane and CO₂. Formation of sulfide or of butane were not detected in biotic controls in the absence of butane. We conclude that the observed enrichment of butane in $^{13}$C demonstrates steady fluxes of $^{13}$CO₂ to the butane pool via reverse AOB reactions.

To quantify the ratio of the AOB back flux to the net AOB process, we developed an empirical mass-balance isotope model that accounts for the contribution of both forward and reverse reactions to the observed isotope dynamics. The model returned a bell-shaped profile of the relative contribution of the back flux, with the highest proportion of the back flux ranging from 1.5 to 2.5‰ of the net AOB rate

(Fig. 4). This is approximately one order of magnitude lower than the carbon back fluxes reported for AOM and AOE[7,23]. For the initial growth phase, which corresponds to the gradual enrichment of the residual butane pool in $^{13}$C, modelling returned a progressive increase of the back flux extent. The flux of reaction products to substrates is predicted to increase as the free energy change is approaching equilibrium ($\Delta G \rightarrow 0$)[4]. Indeed, energetic modelling showed that the increase in back flux was accompanied by a shift of the AOB free energy towards more positive values, however not approaching thermodynamic equilibrium. This indicated nevertheless that the increase of AOB back flux was driven by a weakening of the net thermodynamic drive for the oxidative direction (Fig. 4). For the late growth phase when the $^{13}$C enrichment of the residual butane pool remained constant, the model returned a gradually diminished back flux even though the $\Delta G$ of the net AOB process continued to slightly shift towards equilibrium (Fig. 4). To test whether this may have been caused by depletion of butane during late growth phases, back flux experiments were conducted under low sulfate concentrations (5 mmol l$^{-1}$), conditions known to foster higher back fluxes during AOM[24]. In these assays depletion of sulfate was accompanied by consumption of only about half of the added butane (6 mmol l$^{-1}$) (Fig. 5A). Nevertheless, we measured only a minor $^{13}$C enrichment of the residual butane pool, which

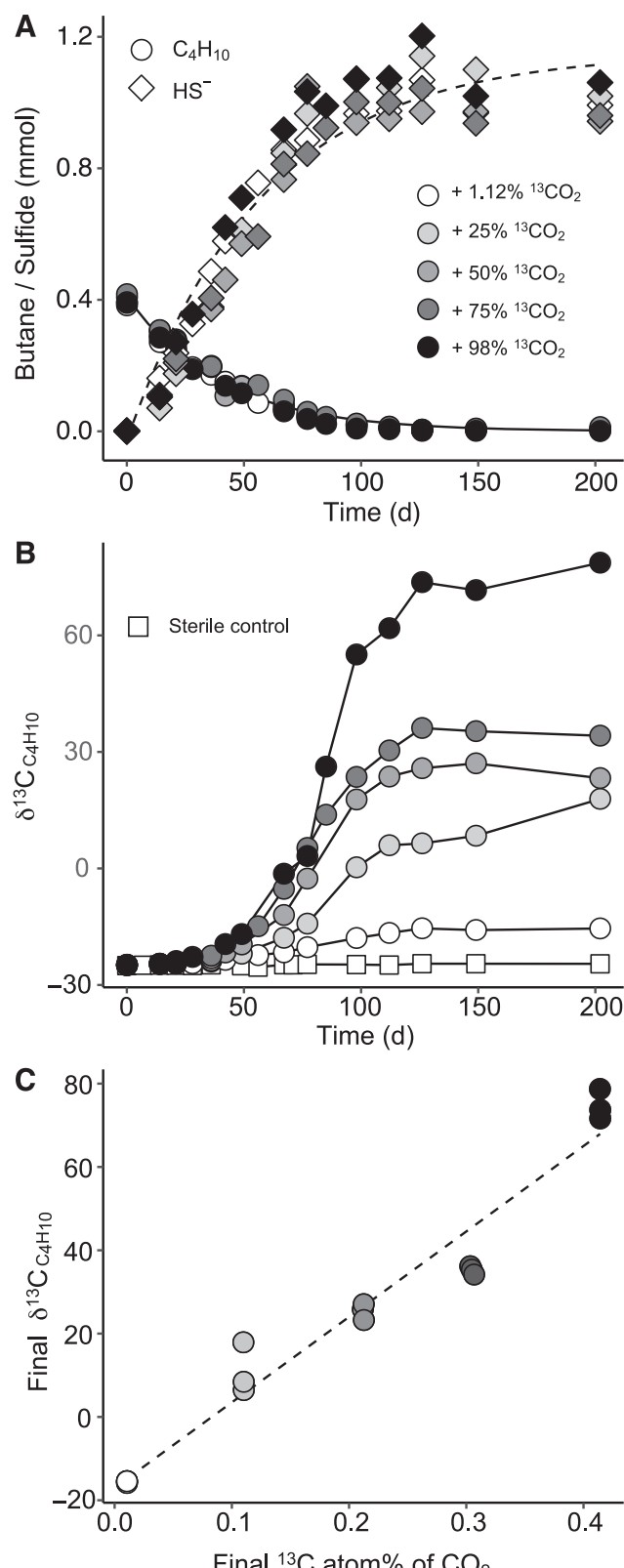

**Fig. 3 | Experimental data for the AOB cultures incubated with increasing fractions of ¹³C-labeled DIC. A** Consumption of butane (circles) and formation of sulfide (diamonds) in cultures supplied with ¹³C-DIC ranging from 1.12 to 98%. **B** Development of $\delta^{13}C$ values (in ‰) in the residual butane pools (circles); color shades represent the same ¹³C-DIC labeling ratios as in panel A; $\delta^{13}C$ of butane in sterile controls showed no variations during the same incubation time (squares). **C** Correlation of $\delta^{13}C$ of residual butane pools with the ¹³C abundance in the DIC pool during stationary phases. Source data are provided as a Source Data file.

the still large residual pool of unlabeled butane (approx. 24 mmol l⁻¹). Assays with limiting amounts of butane (3 mmol l⁻¹) led to higher ¹³C enrichment of the residual butane pool, with $\delta^{13}C$ reaching over +100‰ and again remaining constant once butane was depleted (Fig. 5D, E). Modelling showed reduced AOB back fluxes under limitations of either sulfate or butane, corresponding to up to 0.5‰ and 0.8‰ of the net AOB, respectively (Fig. 5C, F).

## Discussion

The back flux assay results allow two main observations. First, the extent of the AOB back reaction in *Ca.* Syntrophoarchaeum cultures was low. Lower back fluxes are expected for the anaerobic oxidation of higher alkanes, as the oxidation reaction becomes more exergonic with increasing alkane chain length (Supplementary Note 7 and Supplementary Tables 4 and 5). We consider that back fluxes as measured here are likely to have a minor impact on isotopic signatures of butane in gas reservoir samples. Second, *Ca.* Syntrophoarchaeum cultures apparently do not catalyze an isotope exchange between $CO_2$ and butane in the absence of net AOB, unlike AOM, and potentially AOE, where forward and back fluxes continue even in the absence of net oxidation[7,24].

We reasoned that the observed back flux dynamics displayed by *Ca.* Syntrophoarchaeum reflected integrated outcomes of energetics and kinetics effects underlying the AOB pathway. According to the flux force theorem[25,26], the shift of AOB free energy towards more positive values (shift trend towards 0) predicts a monotonic increase of AOB back flux (Eqs. 19 and 20), but does not explain the observed sharp decrease of the back flux in the later experimental stages (Fig. 4). The latter could be likely explained by kinetic bottlenecks along the pathway, governed by shrinking pools of pathway intermediates and of reducing equivalents, including those mediating the transfer of electrons to the partner SRB. When cultures are under sulfate or butane limitation (for example, at late growth stages when most butane has been oxidized), less energy is available to drive the endergonic AOB reactions, leading to reduced cell capacity to replenish intermediate pools and to regenerate $XH_2$. The resulting intermediates or $XH_2$ shortages limits enzyme kinetics, lowering the magnitude of AOB back fluxes. Particularly affected could be the energy coupled reactions like the conversion of butyl-CoM to butyryl-CoA.

Overall, our results show that the entire AOB pathway is essentially reversible, suggesting that given a supply of electron donors with low redox potential[27] archaea could catalyze the formation of higher alkanes via ACR-based pathways. In an attempt to identify or emulate the potential electron donor, we supplied sulfate-free cultures of *Ca.* Syntrophoarchaeum with $H_2$, sulfide, zero-valent sulfur compounds, and reduced artificial electron carriers (AQDS). These failed to stimulate a reverse AOB pathway, leaving open the question of the identity of $XH_2$. Similarly, the nature of X could not be resolved by metagenomic analyses (Supplementary Note 1). Generally, the archaeal AOAlk pathways are considered to have evolved by metabolic modular extension of AOM, and evolutionary adaptations of MCRs to accommodate substrates bulkier than methane[6,12]. All known constituent metabolic modules, including the Wolfe, Wood-Ljungdahl, and beta-oxidation pathways, are known to be reversible[28,29]. The reactions converting acyl-CoA to alkyl-CoM are presently unknown

stopped as soon as sulfate was depleted (Fig. 5B). For reference, back flux assays were done under variable starting butane concentrations (Fig. 5D, E) and high sulfate (28 mmol l⁻¹). Addition of excess butane (30 mmol l⁻¹) led to high sulfate reduction rates, which stopped once the cultures accumulated about 20 mmol l⁻¹ sulfide. Under these conditions we measured only a marginal enrichment in ¹³C-butane ($\delta^{13}C$ shift from approx. −25‰ to −24‰), most likely due to label dilution in

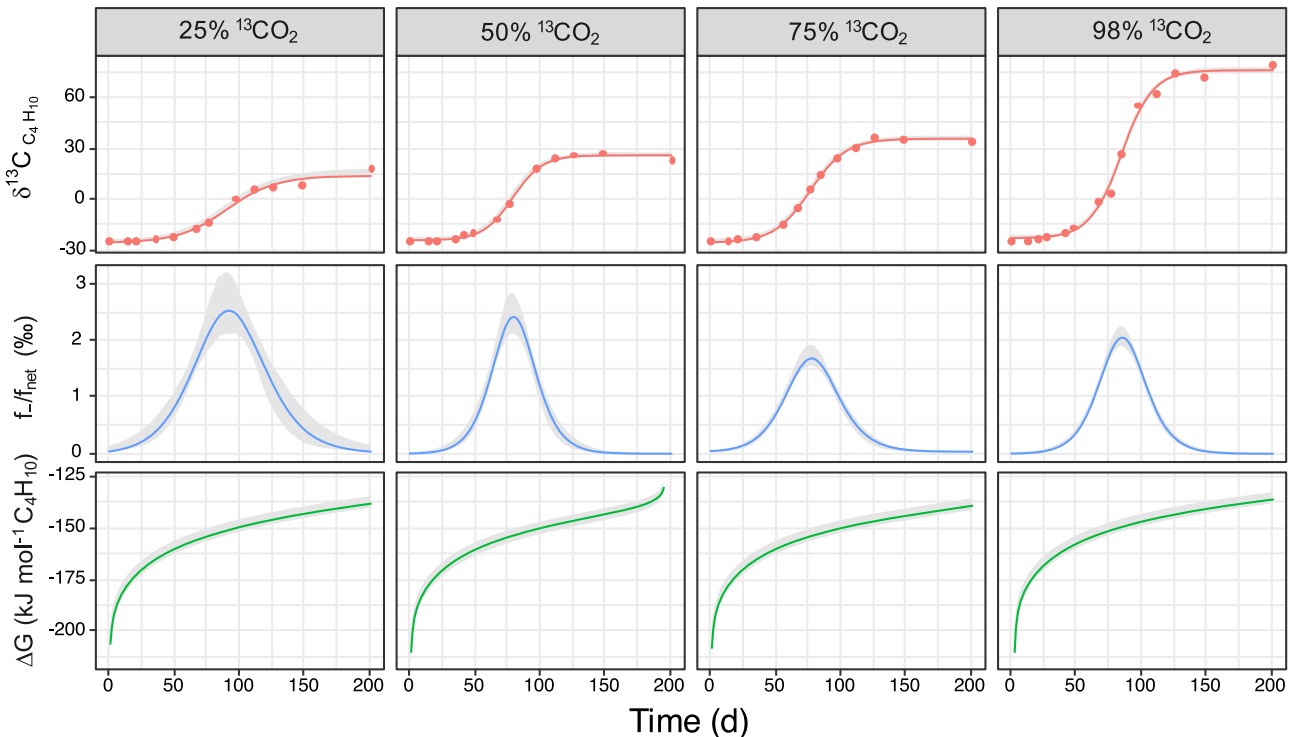

**Fig. 4 | Model output for the dynamics of δ¹³C_butane, back flux and free energy change during AOB under variable ¹³C/¹²C of CO₂.** The upper row shows the time-dependent model fitting (lines) of experimental data (circles) showing the evolution of δ¹³C in residual butane pools. The middle row shows that the contribution of the modelled back flux ($f_-$) to the net AOB process ($f_{net}$) followed similar trends under variable labeling conditions. The bottom row shows that the total free energy change of AOB ($\Delta G$) shifts along the depletion of butane and accumulation of $CO_2$ throughout the incubation. The shading shows the 95% confidence interval of the estimated values. Source data are provided as a Source Data file.

(Supplementary Note 2 and Supplementary Data 1-5). In a hypothetical alkanogenesis scenario, one of the critical steps would be the release of alkane from alkyl-CoM. For this we calculated $\Delta G^{0'}$ ranging from $-19 \pm 10$ kJ mol⁻¹ (ethyl-CoM → ethane) to $-16.9 \pm 10$ kJ mol⁻¹ (butyl-CoM → butane), comparable with the release of methane from methyl-CoM ($-30 \pm 10$ kJ mol⁻¹, Supplementary Note 8). Like in methanogenesis, energy conservation during the hypothetical alkanogenesis could be for example coupled to CoM alkylation and cycling of the CoM-S-S-CoB heterodisulfide[30].

An ACR-dependent alkanogenesis offers an alternative microbiological and mechanistic explanation to the proposed biological origin of $C_{2+}$ volatile alkanes. Ethane and propane depleted in ¹³C relative to thermogenic gasses have been frequently detected in marine gas hydrates, sedimentary basins, and deep marine sediments, often in deeply buried sediments isolated from gas reservoirs[31–35]. In addition, formation of ethane has been observed in sediment slurries, with experimental evidence suggesting an involvement of archaea[35]. To date, the involved microorganisms and underlying biochemical mechanisms have not been identified. Here we propose that such alkanes are formed by archaea harboring ACR-dependent pathways. Moreover, our results expand the range of $C_{2+}$ alkanes that could be biologically formed to include butane. Butane is less abundant in biogenic gas samples, and reports of its isotopic composition are scarce[36]. To test if *Ca.* Syntrophoarchaeum and other alkane-oxidizing archaea (ALOX)[37] occur in environments harboring traces of volatile alkanes or ¹³C-depleted alkanes, we performed a global taxonomy survey of ACR-encoding archaea and of AcrA sequences. We retrieved *Ca.* Syntrophoarchaeum primarily from hot springs, cold seeps, and hydrothermal vents, while other ACR-encoding genera, like Bathyarchaeota[38], are apparently found in a broader range of aquatic and sedimentary ecosystems (Fig. 6). The environmental distribution of AcrA fragments followed closely the

biogeographic patterns of ACR-encoding genera (Fig. 6). Although in most cases the presence of ALOX could not be directly correlated with alkane geochemistry, environments like hot springs apparently host both ACR genes (Fig. 6)[11,39] and traces of $C_{2+}$ alkanes including butane[40]. Such energy-limited chemolithotrophic geothermal settings may be hot spots of both ALOX archaea and of genuine alkanogenic archaea in nature.

## Methods

### Experimental design
The experiments were carried out to demonstrate the back flux of carbon from $CO_2$ to butane pool during net AOB. The back reaction of AOB was measured by isotopically labeling the product ($CO_2$) and measuring the appearance of label in the diminishing substrate pool (butane pool). As previously described, sediment-free thermophilic AOB enrichment culture Butane50 was used for back flux experiments[8]. The Butane50 culture was obtained from a Guaymas Basin vent area (27° 00.437′ N, 11° 24.548′ W; 2000 m water depth). The culture was dominated by a butane-oxidizing archaeon, *Ca.* Syntropharchaeum butanivorans, which forms aggregates with a sulfate-reducing bacterium closely related to *Candidatus* Desulfofervidus auxilii.

### Incubation conditions for the back flux experiment
To control the amount of ¹³C label added to cultures, experiments were conducted in anoxic Tris-HCl buffered artificial seawater (ASW) medium[41,42]. The Tris-HCl buffered ASW medium had the same composition as ASW medium, with the exception that the $CO_2$/bicarbonate-buffer system was replaced by Tris-HCl (final concentration: 30 mM; pH = 7.0). The headspace of Tris-HCl buffered ASW medium was purged with 100% $N_2$, contrasting with a mixture of $N_2$ and $CO_2$ (9:1 by volume) in the atmosphere of $CO_2$/bicarbonate buffered ASW

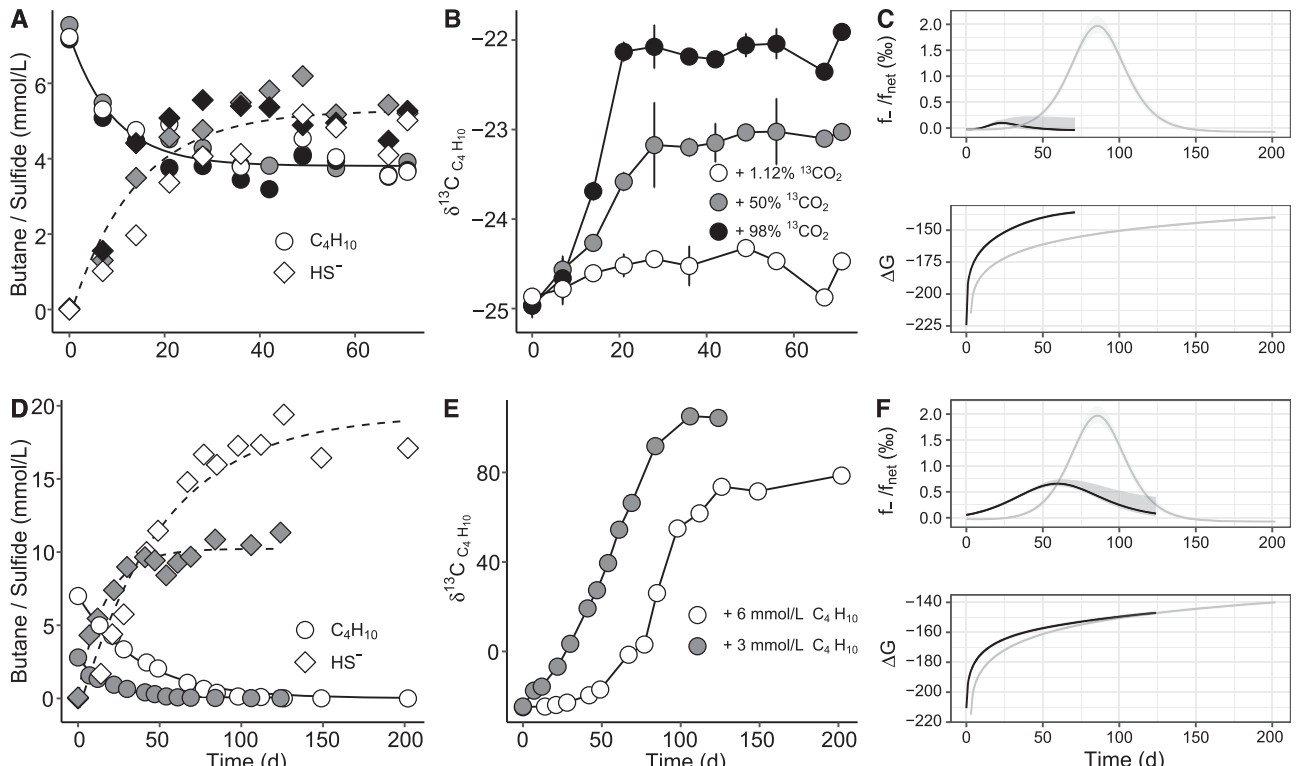

**Fig. 5 | Back flux under sulfate limitation or with variable starting butane concentrations. A, D** Temporal evolution of measured concentrations of butane ($C_4H_{10}$, circles) and sulfide ($HS^-$, diamonds), along with trend lines fitted using the first-order dynamic model. Gray shades indicate measurements from low-sulfate incubations (5 mmol/L sulfate) with three different starting [13]C-bicarbonate concentrations (**A**; 1.12%, 50%, and 98%), or from incubations with two different initial butane concentrations (**D**; 3 mmol l$^{-1}$ vs. 6 mmol l$^{-1}$). **B, E** Development of isotope compositions of butane ($\delta^{13}C_{C_4H_{10}}$). Color shades represent the same incubation conditions as in **A**; Error bars in **B**, **E**, are standard deviations of technical replicates ($n = 3$). **C, F** Ratio of back flux relative to the net AOB rate ($f_-/f_{net}$) and Gibbs free energy change of the overall AOB reaction ($\Delta G$) estimated from measured concentrations and/or isotope compositions. Black lines show calculated values for low-sulfate assays with 50% [13]C-bicarbonate (**C**) and for low-butane assays (**F**). The $f_-/f_{net}$ and $\Delta G$ calculated for the assay with 6 mmol l$^{-1}$ $C_4H_{10}$ 98% [13]C-bicarbonate, and 28 mmol l$^{-1}$ sulfate are shown for comparison (grey lines in **C**, **F**). The shaded envelopes in (**C**, **F**) indicate the 95% confidence interval of estimated values. Source data are provided as a Source Data file.

medium. Growth of the Butane50 culture in Tris-buffered ASW medium was tested beforehand and found to proceed at similar rates as growth in $CO_2$/bicarbonate-buffered medium.

Prior to establishment of back flux experiments, fully grown, aggregate-rich Butane50 cultures were rendered bicarbonate/$CO_2$ free, by exchanging the overlying medium two times with Tris-HCl buffered ASW medium. Replicate cultures ($n = 5$) were set up in 120 ml serum bottles containing a 20 ml aliquot of re-suspended aggregates and 40 ml of Tris-HCl buffered ASW medium. Serum bottles were closed with butyl rubber stoppers (total volume = 118 ml after closing) and flushed with pure $N_2$, leaving 58 ml of headspace. Defined amount of [13]C-labelled bicarbonate (98 atom% [13]C; Sigma Aldrich) and/or unlabeled (natural [13]C abundance) bicarbonate (Sigma Aldrich) were injected into the serum bottles, to reach a final bicarbonate concentration of 18 mM with defined [13]C atom% (i.e., 1.12%, 25%, 50%, 75%, and 98%). Additional replicate assays were set up for natural abundance bicarbonate ([13]C = 1.12%; $n = 2$), and for 98% [13]C bicarbonate ($n = 2$). Unlabeled butane (9 ml) was introduced anaerobically using gas-tight syringes. As abiotic controls, sterile ASW with 9 ml butane and 98 atom% [13]C bicarbonate were prepared ($n = 2$). Additional controls included assays with cells incubated without butane (substrate-free controls; $n = 4$). To probe carbon back flux under sulfate limited concentration, parallel cultures were set up in Tris-HCl buffered ASW medium with reduced sulfate concentration (5 mM; $n = 3$). To test the impact of substrate concentration on the extent of carbon back flux, experiments were done with different starting butane concentrations (4.5, 9, and 58 ml; $n = 3$). All manipulations of Butane50 cultures were

performed under an $N_2$ atmosphere. Cultures were incubated at 50 °C with continuous shaking (100 rpm) in the dark.

**Probe design.** The 16S rRNA gene sequences of *Ca.* Syntrophoarchaeum butanivorans and *Ca.* S. caldarius were imported and aligned to the SILVA database (release 111)[43], using the ARB software package (version 6.0.6)[44]. Sequence-specific oligonucleotide probes and helpers were designed in ARB using the Probe Desing tool using the default parameters: length of probe, 18 nucleotides; temperature, 30–100 °C; GC-content, 50–100%; maximum nongroup hits, 0. Returned probes were selected based on predicted fluorescence intensity signals according to the 16S rRNA mapping in fluorescence in situ hybridization assays[45]. Hybridization efficiency was further evaluated in silico using mathfish[46]. Probe specificity was checked against the SILVA database and the Ribosomal Database Project[47]. Selected probes coupled with horseradish peroxidase (HRP), helpers, and competitor oligonucleotides (Supplementary Table 1) were purchased from Biomers (biomers.net, Ulm, Germany). Probe stringency was experimentally tested in Catalyzed Reporter Deposition Fluorescence In Situ Hybridization assays (CARD-FISH) of the Butane50 culture with increasing formamide (FA) concentrations from 0 to 60% at 10% increments and refined at FA concentrations ranging from 20 to 40% at 5% increments. The stringency of the FA concentration was chosen based on microscopic inspection and evaluation of the fluorescence signals using the same exposure time for all samples (Supplementary Table 1). As controls we used the probes EUB338I-III, ARCH915, and the nonsense probe NON338 (Supplementary Table 1).

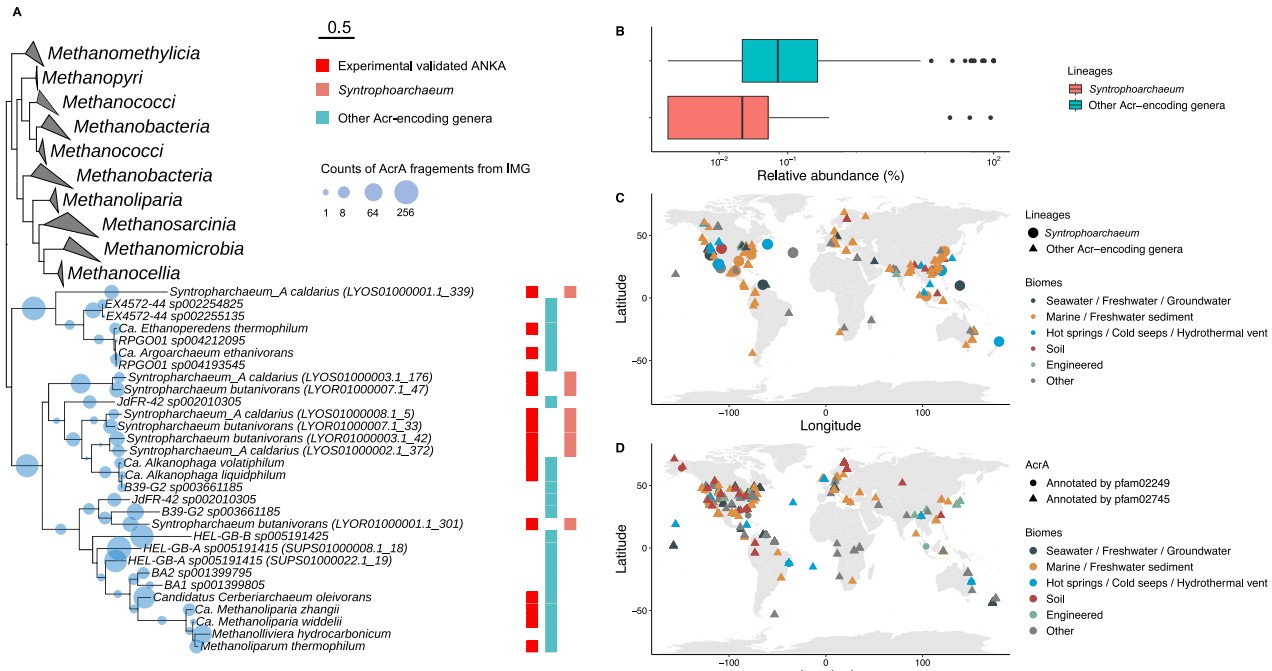

**Fig. 6 | Global biogeography of ACR-encoding archaea and of AcrA sequences. A** Maximum likelihood tree of AcrA protein sequences. AcrA's from experimentally validated anaerobic multicarbon alkane oxidizing archaea are highlighted with red squares. AcrA's from *Ca*. Syntrophoarchaeum and other ACR-encoding genera are marked with red and cyan squares. The transparent circles reflect the number of AcrA fragments from IMG/M database that are placed to the corresponding branches by RAxML EPA. Clades of methanogenic archaea are collapsed for visualization purposes. Scale bar = substitutions per site. **B** Boxplot of relative abundance of *Ca*. Syntrophoarchaeum (red) and other ACR-encoding genera (cyan) shown in the AcrA phylogeny. **C** Biogeographic distribution of *Ca*. Syntrophoarchaeum (circles) and other ACR-encoding genera (triangles) across diverse biomes. **D** Biogeographic distribution of AcrA sequences retrieved from environmental samples. Sequences were retrieved from assembled metagenomes in the JGI IMG/M database. The phylogenetic placement of fragments is shown in **A**. The fragments that are annotated with C-terminus (PF02249) and N-terminus (PF02745) domains of McrA are shown with circles and triangles, respectively. Source data are provided as a Source Data file.

**Catalyzed reporter deposition fluorescence in situ hybridization.**
For CARD-FISH, 1-ml samples were collected from three independent Butane50 cultures using $N_2$-flushed syringes, and immediately transferred in 1 ml of a 4% paraformaldehyde solution (PFA) (electron microscopy grade; Electron Microscopy Sciences) in 1× PBS. Fixation was done overnight (17 h) at 4 °C with. Aggregates were collected by centrifugation, washed three times with 1× PBS, suspended in 80% ethanol (v/v in water), and stored at −20 °C. For CARD-FISH and imaging of native aggregates, subsamples in PFA were filtered on GTTP type 0.22 μm pore size filters (Millipore) at the end of the fixation period. Additionally, to evaluate the stringency of the FA concentrations, and for cell counting, aggregates were disrupted by sonication. For this, PFA-fixed subsamples were treated by mild sonication (probe MS73, Sonopuls UW70, Bandelin, Berlin, Germany) at an amplitude of 20% for 30 s (three 10 s cycles with 10 s break) in a pulsation mode (0.1 s) on ice, at an immersion depth of 20 mm. Following sonication, disrupted samples were filtered onto polycarbonate GTTP 0.22 μm pore size filters.

Hybridizations were performed as described elsewhere[48]. Filters with native aggregates or dispersed cells were coated with 0.2% low melting point agarose kept at 48 °C (Biozym Scientific GmbH) using a spin coater type SCI-40 (LOT-QuantumDesign GmbH) at 50 rotations per minute to ensure homogenous distribution of agarose on the sample surface. Bacteria were permeabilized with lysozyme (10 mg ml⁻¹ in 0.05 M EDTA, pH 8.0; 0.1 M Tris-HCl, pH 7.5) for 30 min at 37 °C. Archaea were permeabilized by incubation with 0.1 M HCl for 1 min, followed by proteinase K (15 μg ml⁻¹) for 5 min at room temperature. Endogenous peroxidases were inactivated by incubation with 0.15% $H_2O_2$ in absolute methanol (30 min, room temperature). The filters were hybridized for 3 h at 46 °C in standard hybridization buffer[48]. The concentration of the HRP-probes was 0.17 ng ml⁻¹. All hybridizations were

performed at 46 °C. Following hybridization, the hybridized filters were incubated for 15 min at 48 °C in prewarmed washing buffer. CARD was done by incubating the filters for 15 min at 46 °C in the dark in standard amplification buffer[48] containing either 1 μg ml⁻¹ Alexa Fluor™ 488 or Alexa Fluor™ 594 labeled tyramides. Tyramide conjugates of Alexa Fluor™ 488 and Alexa Fluor™ 594 were purchased from Thermo Fisher Scientific and used according with the manufacturer's specifications. The hybridized cells were further stained for 10 min in the dark with 1 μg ml⁻¹ of 4′,6′-diamidino-2-phenylindol (DAPI) prepared in deionized water. For fluorescence microscopy evaluation, the filters were embedded in a 4:1 (v/v) mixture of low fluorescence glycerol mountant (Citifluor AF1, Citifluor Ldt., London, UK) and mounting fluid Vecta Shield (Vecta Laboratories, Burlingame, CA USA). Hybridizations were evaluated by fluorescence microscopy using an Axio Imager.Z2 microscope (Carl Zeiss) with a 100× Plan-Apochromat oil objective (Numerical Aperture, NA = 1.4) and filter sets for DAPI, Alexa594 and Alexa488. Evaluation and imaging of native aggregates was done with a confocal laser scanning microscope Zeiss LSM 980 with Airyscan 2 (Zeiss, Germany), using a 63× Plan-Apochromat oil objective, NA = 1.4.

Dual hybridizations were performed consecutively using the following combinations of HRP-labelled oligonucleotide probes: ARCH915 + EUB338, pSBU699 + ARCH915, pSBU699 + pSCAL871, or pSCAL871 + ARCH915. Signal discrimination in dual hybridizations was done using the tandem Alexa Fluor 594 for one of the probes and Alexa Fluor 488 for the second probe. For the dual CARD-FISH procedures the cell-wall permeabilization was done as described above for single hybridizations but sequentially for bacteria and archaea, respectively, and prior to the first hybridization. Before the second hybridization, the HRP introduced in the first hybridization was inactivated by incubation for 10 min at room temperature with 3% $H_2O_2$. Standard mounting and epifluorescence microscopy was used for visualization.

**Amplicon sequencing.** DNA was extracted from two independent Butane50 cultures as previously described[6]. An additional step of cell lysis using a bead-based benchtop homogenizer (Bead Mill MAX, VWR; 5 cycles of 50 s reciprocal shaking at 2000 r.p.m., with 15 s pause) was done before the freeze-thaw cycle. The V3–V4 region of the bacterial 16S rRNA gene was amplified using the universal prokaryotic primers 314 F and 805 R[6,49]. The paired end sequencing ($2 \times 300$ bp) was done on an Illumina MiSeq desktop sequencer using the V3 sequencing kit (Illumina, Inc., San Diego, CA, USA).

All sequence analyses were performed in R v. 4.4.1. Barcodes and primers were trimmed using cutadapt v. 0.2.0[50]. Amplicon sequence variant (ASV) assignment was performed with the R package 'DADA2' v. 1.28.0[51]. This included error correction, ASV calling, chimera removal, and taxonomic classification according to the Silva SSU reference database nr. 138[43]. PCR negatives were used for decontaminating the data using the R package Decontam v. 1.20.0[52]. Sequencing data has been submitted to NCBI under the BioProject ID PRJNA1134092.

## Chemical analyses of sulfide and butane
Sulfide concentrations were determined using a previously described colorimetric method[53]. Briefly, 0.1 ml medium was sampled from cultures with $N_2$-flushed syringes and mixed with 4 ml of an acidified copper sulfate (5 mM) solution. The formed colloidal copper sulfide was quantified photometrically at 480 nm. To monitor butane consumption over time, 0.2 ml of headspace gas was withdrawn from cultures using gas-tight syringes flushed with $N_2$. The butane concentrations were measured using a gas chromatograph (Shimadzu GC-14B), equipped with a flame ionization detector. Compound separation was achieved using a Supel-Q PLOT column ($30 \, m \times 0.53 \, mm$, 30 μm film thickness; Supelco, Bellefonte, USA), with $N_2$ as carrier gas at a flow rate of 3 ml min⁻¹. The oven was maintained at 140 °C, with the injection and detection temperature kept constant at 150 °C and 280 °C, respectively. Butane concentrations were calculated based on external calibration curves and reported as the mean value of duplicate measurements (technical replicates).

## Stable isotope analysis
To measure the isotope composition of butane, 0.2 ml of headspace gas was collected anaerobically in gas-tight sealed serum vials (12 ml) containing 1 ml of saturated NaCl solution. The vials were stored upside down to prevent adsorption of butane to stoppers. Carbon isotope analyses were done on a Thermo Scientific MAT 253 isotope-ratio mass spectrometer interfaced with a Trace 1310 GC system via a GC-IsoLink and a ConFlo IV interface (all Thermo Scientific). The combustion reactor was maintained at 1000 °C. Sample separation was done with a PoraBOND Q column ($50 \, m \times 0.32 \, mm \times 5 \, μm$; Agilent Technology) using a constant helium carrier gas flow of 2 mL min⁻¹. The oven temperature was initially held at 40 °C for 5 min, and increased to 250 °C at a rate of 10 °C min⁻¹, with a final hold at 250 °C for 5 min. The injector temperature was kept at 250 °C and a split ratio of 1:20, 1:10, or 1:5 was used depending on the concentration of butane in the samples. All samples were analyzed at least three times. Stable carbon isotopic ratios were reported as δ notation ($\delta^{13}C$) in parts per thousand (‰) relative to the international standard scale (VPDB). The isotopic fractionation factor associated with the forward AOB flux ($\alpha_+$) was calculated using the logarithmic form of the Rayleigh equation as previously described (Supplementary Note 6;[22]).

## Thermodynamic analyses of anaerobic oxidation of butane (AOB) and its reverse pathway (rAOB)
The AOB catalyzed by *Ca*. S. butanivorans, generalized as $C_4H_{10} + 13X \rightarrow 4CO_2 + 13XH_2$ (with $XH_2/X$ being a generic two-electron carrier) consists of 17 intermediate steps (Table 1), which we grouped into four metabolic modules: butane activation to butyl-CoM (Reactions 1–2), conversion of butyl-CoM to butyryl-CoA (Reaction 3), beta-oxidation of butyryl-CoA to acetyl-CoA (Reactions 4–8), and the reverse Wood-Ljungdahl pathway (Reactions 9–14). Reactions 15–17 are energy conservation reactions.

To enable thermodynamic analysis of the pathway, assumptions were made regarding the redox potential of the couple $XH_2/X$, the unresolved catabolic reactions, and energy-transduction mechanisms. Most assumptions were supported by a refined analysis of *Ca*. S. butanivorans genome:

1. The electron carrier ($XH_2/X$) has a standard redox potential of $Ex$ in volt (V). Two different $Ex$ values were selected to represent alternative hypotheses on the syntrophic metabolism between archaea and sulfate-reducing bacteria (SRB)[8,54,55]. These encompass pili- and cytochrome-based direct interspecies electron transfer, and indirect electron transfer using diffusible redox compounds (e.g., $H_2$). In the first scenario, we considered that the syntrophic SRB accepts reducing equivalents from butane-oxidizing archaea via nanowires; the value of Ex was thereby set to the average potential of sulfate reduction ($E^{0'} = -0.220 \, V$). In the latter case, $Ex$ was equated to the redox potential of $H_2$ at pH = 7.0 ($E^{0'} = -0.414 \, V$; biochemical standard) to represent the potential interspecies transfer of reducing equivalents via $H_2$.

2. Recycling of CoM-S-S-CoB involves an electron-confurcating mechanism coupling the reduction of two $F_{420}H_2$ with the oxidation of coenzyme B (CoB-SH), coenzyme M (CoM-SH), and ferredoxin (Reaction 2 in Table 1). This assumption is supported by the presence of genes for a cytoplasmic complex composed of a heterodisulfide reductase (HdrABC) and a $F_{420}$ hydrogenase subunit B (FrhB) in the genome of *Ca*. S. butanivorans (Supplementary Note 9 and Supplementary Table 2). The HdrABC-FrhB complex couples the exergonic reduction of $F_{420}$ ($E^{0'} = -0.340 \, V$) with ferredoxin ($E^{0'} = -0.500 \, V$) to the endergonic reduction of $F_{420}$ with CoB-SH + CoM-SH ($E^{0'} = -0.143 \, V$) via a flavin-based electron confurcation mechanism, as previously suggested for methane-oxidizing ANME-1 and *Methanoperedenaceae*[56,57].

3. The unsolved process converting butyl-CoM to butyryl-CoA was generalized as butyl-S-CoM + CoA + $H_2O$ + 2X → butyryl-CoA + HS-CoM + $2XH_2$ (Reaction 3 in Table 1). The simplification is justified considering that the transformation of one butyl to one butyryl moiety releases four electrons, which are ultimately channeled to the terminal electron acceptor X, forming two $XH_2$.

4. Menaquione (MQ) is the membrane-bound soluble electron carrier of *Ca*. S. butanivorans (Reactions 8 and 15–17 in Table 1). In support of this, the futalosine pathway for MQ biosynthesis was predicted in the genome of *Ca*. S. butanivorans, similar to the *Methanoperedenaceae* and ANME-1 (*mqnABCDEX* in Supplementary Table 2).

5. The NADH:quinone oxidoreductase (Nuo) from the genome of *Ca*. S. butanivorans acts as an ion-translocating ferredoxin:quinone oxidoreductase. The putative Nuo complex lacks the NADH-oxidizing subunits NuoEFG, suggesting it is incapable of oxidizing NADH. Such a truncated Nuo complex has been supposed to oxidize ferredoxin while reducing MQ and generating a chemiosmotic ion gradient[58,59]. Under this scenario, the NADH yielded from beta-oxidation (Reactions 4 and 6) and the Wood-Ljungdahl pathway (Reaction 10) is oxidized by a non-proton-translocating, single-subunit NDH predicted from *Syntrophoarchaeum* genome (Reaction 8 in Table 1, Supplementary Table 2).

6. Electron transfer from the membrane-bound MQ to the terminal electron acceptor X involves an ion-translocating membrane protein (Reaction 17). The oxidation of reduced MQ releases electrons with far higher potential ($MQH_2/MQ$: $E^{0'} = -0.080 \, V$) compared to X ($E^{0'} \leq -0.220 \, V$). A plausible mechanism achieving this involves exploitation of the PMF, where electrons move upward the energy gradient by consuming PMF. This energy-driven reverse electron flow constitutes a prevalent phenomenon in syntrophic microorganisms[60].

For each AOB reaction the standard Gibbs free energy changes ($\Delta G^{0'}$, considering 298 K, pH 7.0, and an activity of 1 for products and substrates) were collected from literature or calculated using standard Gibbs free energy of formation ($\Delta G_f^{0'}$) or standard redox potentials ($E^{0'}$). $\Delta G_f^{0'}$ and $E^{0'}$ refer to biological standard conditions with 1 M concentration of all reactants at pH 7.0. For the reactions with no experimentally validated thermodynamic data, $\Delta G^{0'}$ was predicted using the online tool Equilibrator[61] considering pH = 7.0, pMg = 0, and ionic strength = 0 M. Derivation of $\Delta G^{0'}$ for each step can be found in the Supplementary Data 6 and 7. The $\Delta G^{0'}$ for each step of reverse AOB (rAOB) was obtained by changing the $\Delta G^{0'}$ sign of the corresponding AOB step.

The feasibility of each reaction direction (AOB/rAOB) was analyzed by calculating quasi-equilibrium concentrations of intermediates under thermodynamic constraints. A pathway was deemed unfeasible when quasi-equilibrium concentrations of intermediates deviate outside physiological concentration ranges. Specifically, extremely low metabolite concentrations (i.e., <1 µM) would impose bottlenecks on enzymatic kinetics, rendering the catabolic pathway non-functional. Conversely, very high metabolite concentrations (i.e., >10 mM) were deemed physiological unfeasible, constrained by cell biomass, cell volume, and solubility. In this study, a low threshold of 1 µM and a high threshold of 10 mM were selected, consistent with the established thermodynamic framework[19].

To calculate the quasi-equilibrium concentrations, thermodynamic equilibrium ($\Delta G = 0$) was assumed for most AOB/rAOB reactions, except those thought to be coupled with biological energy transduction. Highly exergonic reactions ($\Delta G \ll 0$) enable energy conservation via build-up of PMF, whereas endergonic reactions ($\Delta G > 0$) require energy investment in the form of PMF to overcome thermodynamic barriers. This PMF-based energy transduction mechanism allows specific steps within the pathway to operate departing from thermodynamic equilibrium ($\Delta G \neq 0$), with the actual free energy equating to the quantum of energy required to translocate one proton across the membrane ($\Delta\mu_{H+} = -20$ kJ mol$^{-1}$)[19]. For a few highly exergonic reactions, energy dissipation was assumed to avoid bottlenecks otherwise occurring under thermodynamic equilibrium[19]. To calculate the actual free energy values ($\Delta G$) of each AOB/rAOB reactions, we established a set of linear equations (Eq. 1), with log concentrations of each intermediate as unknown variables. The solutions of the linear equations provided quasi-equilibrium metabolite concentrations, allowing to assess the feasibility of AOB and rAOB.

$$\Delta G = \Delta G^{0'} + RT \ln \prod_i a_i^{s_i} = \Delta G^{0'} + RT \sum_i s_i \ln a_i \qquad (1)$$

where $\Delta G^{0'}$ is the standard free energy change; $s_i$ and $a_i$ are the stoichiometry coefficient and activity of metabolite $i$, respectively. Activities of metabolites were assumed to be equal to actual concentrations neglecting ionic strengths.

### Isotope mass balance model for the anaerobic oxidation of $n$-alkane (AOAlk)

We developed an isotope mass balance model to quantify the extent of back flux during net anaerobic oxidation of volatile $C_{2+}$ alkanes (AOAlk). The model is modified from a previous model[4] to account for back flux of $C_{2+}$ alkanes, and to enable data evaluation for $^{13}C$-based labelling experiments versus $^{14}C$-based radiotracer experiments. All terms used for the model development are listed in Supplementary Table 6. The modified model simulates the evolution of carbon isotope composition in the $n$-alkane pool during AOAlk and estimates the ratio of back flux relative to the net AOAlk rate by fitting model predictions to experimental measurements. Unlike the previous study[4], we considered the contribution of both backward ($f_-$) and forward fluxes

($f_+$) of AOAlk to the change of carbon isotope in the $n$-alkane pool with an infinitesimal time span (d$t$). The reverse reaction (formation of $n$-alkane) transfers $^{13}C$ from the labelled $CO_2$ pool to the $n$-alkane pool, leading to progressive isotope enrichment (Supplementary Fig. 6; Eq. 2). In parallel, the forward reaction (oxidation of $n$-alkane) takes away $^{13}C$ from the $n$-alkane pool, resulting in label dilution (Supplementary Fig. 6; Eq. 2). The rate of label transfer in each direction depends on the $f_-$ and $f_+$, kinetic isotope fractionation factors associated with each flux ($\alpha_-$ and $\alpha_+$), and the proportion of label in the $CO_2$ and $n$-alkane pools ($\left[^{13}C_{CO_2}\right]/[CO_2]$ and $\left[^{13}C_{Alk_n}\right]/n[Alk_n]$).

$$\frac{d\left[^{13}C_{Alk_n}\right]}{dt} = f_- \alpha_- \frac{\left[^{13}C_{CO_2}\right]}{[CO_2]} - nf_+ \alpha_+ \frac{\left[^{13}C_{Alk_n}\right]}{n[Alk_n]} \qquad (2)$$

By definition, the observed net disappearance rate of $n$-alkane ($v$) amounts to the net fluxes of AOAlk operating in forward and backward direction (Eqs. 3 and 4).

$$\frac{d[Alk_n]}{dt} = -v \qquad (3)$$

$$v = f_+ - \frac{f_-}{n} \qquad (4)$$

In a closed system, the total amount of carbon atoms and stable isotope atoms ($^{13}C$) remain constant over time (mass conservation) and are equal to the amount of carbon and label initially added to the system (Eqs. 5 and 6). Anaerobic alkane oxidizing consortia of archaea-SRB or SRB-dominated enriched or pure cultures are slow growing, with low biomass yield[8,15]. Mass balance calculations showed that during the labeling experiments the amount of butane assimilated by the Butane50 culture was on average ($n = 5$ replicates) 4% of the initially added butane (Supplementary Table 3), with minimal impact on the isotope mass balance model (Supplementary Table 7).

$$[CO_2] + n[Alk_n] = [CO_2]_0 + n[Alk_n]_0 = [C]_0 \qquad (5)$$

$$\left[^{13}C_{CO_2}\right] + \left[^{13}C_{Alk_n}\right] = \left[^{13}C_{CO_2}\right]_0 + \left[^{13}C_{Alk_n}\right]_0 = \left[^{13}C\right]_0 \qquad (6)$$

$[CO_2]_0$ and $[Alk_n]_0$ are the initial amounts of $CO_2$ and $n$-alkane, respectively; $[^{13}C_{CO_2}]_0$ and $[^{13}C_{Alk_n}]_0$ are the initial amounts of $^{13}C$ in the $CO_2$ and $n$-alkane pools, respectively; $[C]_0$ and $[^{13}C]_0$ are the total amounts of carbon and $^{13}C$ in the system, respectively.

The extent of the back flux ($\mu$) is defined as the ratio of back flux ($f_-$) relative to net AOAlk rate (Eq. 7). It transforms $f_-$ and $f_+$ as a function of $v$ (Eq. 7).

$$\mu = \frac{f_-}{nv} \Longrightarrow f_- = \mu nv, f_+ = (1+\mu)v \qquad (7)$$

Equation 2 becomes Eq. 8 by denoting $^{13}F_{Alk_n}$ and $^{13}F_{CO_2}$ as fractional abundance of $^{13}C$ in the $n$-alkane and $CO_2$ pool, and by substituting $f_-$ and $f_+$ with $v$ and $\mu$ using Eq. 7.

$$\frac{d\left[^{13}C_{Alk_n}\right]}{dt} = n\mu v\alpha_- \,^{13}F_{CO_2} - n(1+\mu)v\alpha_+ \,^{13}F_{Alk_n} \qquad (8)$$

### Derivation of AOAlk back flux extent

The overall approach to derive the extent of the back flux ($\mu$ in Eq. 8) from experimental data consists of two steps: (1) approximating the changes in pool size and isotope composition of $n$-alkane over time using empirical dynamic models; (2) substituting the approximated models/equations in

Eq. 8 to yield the extent of back flux during AOAlk (Supplementary Fig. 7). In the first step, we used a first-order reaction model with three parameters ($c_0$, $c_1$, and $\lambda$) to describe the net AOAlk process. It predicts that the amount of $n$-alkane ($[Alk_n]$) is subjected to exponential decay over time (Eq. 9) at a net rate ($v$) proportional to the residual quantity of $n$-alkane (Eq. 10). This first-order behavior is common in many microbial batch incubations and matches closely with our experimental data (Fig. 2). To approximate the evolution of carbon isotope composition in the $n$-alkane pool, we exploited a logistic model with four parameters ($d_0$, $d_1$, $d_2$, $k$ in Eq. 11). It captures the 'S-shaped' dynamics of measured $\delta^{13}C$ of $n$-alkane ($\delta^{13}C_{Alk_n}$ in Fig. 2). By definition, carbon isotope ratio of $n$-alkane ($^{13}R_{Alk_n}$) was converted from approximated $\delta^{13}C$ signature with known $^{13}C{:}^{12}C$ ratio for the standard material PDB ($^{13}R_{Std} = 0.01123720$ in Eq. 12). The absolute amount of $^{13}C$ in the $n$-alkane pool ($\left[^{13}C_{Alk_n}\right]$) was further deduced from isotope ratios $^{13}R_{Alk_n}$ and estimated $n$-alkane amount $[Alk_n]$ (Eq. 13).

$$[Alk_n] = c_0 + (c_1 - c_0)\exp(-\lambda t) \qquad (9)$$

$$v = -\frac{d[Alk_n]}{dt} = \lambda\left([Alk_n] - c_0\right) \qquad (10)$$

$$\delta^{13}C_{Alk_n} = d_0 + (d_1 - d_0)/(1 + \exp(-k(t - d_2))) \qquad (11)$$

$$^{13}R_{Alk_n} = {}^{13}R_{Std}\left(1 + \frac{\delta^{13}C_{Alk_n}}{1000}\right) \qquad (12)$$

$$\left[^{13}C_{Alk_n}\right] = \frac{n\,{}^{13}R_{Alk_n}}{{}^{13}R_{Alk_n} + 1}[Alk_n] \qquad (13)$$

In the second step of the derivation, we replaced time-variant variables ($\frac{d\left[^{13}C_{Alk_n}\right]}{dt}$, $^{13}F_{CO_2}$ and $^{13}F_{Alk_n}$) in Eq. 8 with $[Alk_n]$, $^{13}R_{Alk_n}$, $\left[^{13}C_{Alk_n}\right]$, and $v$ from the first step (Eqs. 9–13). This led to an explicit expression of $\mu$ (Eq. 14).

$$\mu = \frac{\frac{d\left[^{13}C_{Alk_n}\right]}{dt} + nv\alpha_+\,{}^{13}F_{Alk_n}}{nv\alpha_-\,{}^{13}F_{CO_2} - nv\alpha_+\,{}^{13}F_{Alk_n}} \qquad (14)$$

where $\frac{d\left[^{13}C_{Alk_n}\right]}{dt}$ was calculated by taking the first-order derivative of $\left[^{13}C_{Alk_n}\right]$ (Eq. 15); $^{13}F_{CO_2}$ was deduced based on mass conservation of carbon atoms and $^{13}C$ in a closed system (Eq. 16); $^{13}F_{Alk_n}$ was converted from isotope ratio by definition (Eq. 17).

$$\frac{d\left[^{13}C_{Alk_n}\right]}{dt} = \frac{n[Alk_n]}{\left(1 + {}^{13}R_{Alk_n}\right)^2}\frac{{}^{13}R_{Std}}{1000}\frac{(d_1 - d_0)ke^{-k(t-d_2)}}{\left(1 + e^{-k(t-d_2)}\right)^2} - \frac{n\,{}^{13}R_{Alk_n}}{{}^{13}R_{Alk_n} + 1}v \qquad (15)$$

$$^{13}F_{CO_2} = \frac{[^{13}C]_0 - \left[^{13}C_{Alk_n}\right]}{[C]_0 - n[Alk_n]} \qquad (16)$$

$$^{13}F_{Alk_n} = \frac{{}^{13}R_{Alk_n}}{{}^{13}R_{Alk_n} + 1} \qquad (17)$$

As illustrated in Supplementary Fig. 7, the derivation above indicated that the extent of the back flux $\mu$ is based on: (1) parameters in first-order reaction model ($c_0$, $c_1$, $\lambda$) and logistic model ($d_0$, $d_1$, $d_2$, $k$), which can be fitted from experimental data; (2) the initial amount of carbon ($[^{13}C]_0$) and of stable isotope ($[^{13}C]_0$) added to the system; and (3) fractionation factors associated with back ($\alpha_-$) and forward fluxes ($\alpha_+$). In the case of AOB, the number of carbon atoms in butane, $n = 4$; $\alpha_+$ was calculated from the AOB fractionation experiment (incubations with $CO_2$ at natural $^{13}C$ abundance; Supplementary Fig. 5). The value of $\alpha_-$ (fractionation factor associated with biological butane formation from $CO_2$) remained experimentally inaccessible and was thereby set to 1. Note that sensitivity analysis showed that the calculation of $\mu$ is robust to the choice of $\alpha_-$ ranging from 0.90 to 1 (Supplementary Figs. 8 and 9).

## Statistical inference of back flux extent of AOAlk

We established a Bayesian framework for analyzing experimental data to infer the back flux extent of AOAlk ($\mu$) and assess the uncertainty in the inference. We employed a Markov chain Monte Carlo (MCMC) sampler to sample parameters in a first-order reaction model ($c_0$, $c_1$, $\lambda$) and in a logistic model ($d_0$, $d_1$, $d_2$, $k$) from their joint posterior distribution. By running the isotope mass balance model with combinations of sampled parameters (Eq. 17), we generated an ensemble of $\mu$ that converged to the posterior predictive distribution. This distribution naturally accounted for uncertainties of the model parameters that were propagated from experimental measurements.

The MCMC sampling was performed using the *emcee* package in Python 3.9. Uniform distributions over a wide range of permissible parameter space were used as priors (Supplementary Table 8). The likelihood function has the Gaussian form (Eq. 18), which essentially measures model-data mismatches (sum of squared errors). To sample the model parameters from posterior distribution, multiple MCMC chains ($n = 128$) were started in parallel from randomly selected initial values, with each chain running for 5000 cycles. The convergence of MCMC chains were examined using *autocorr.integrated_time()* function in *emcee* package and Gelman–Rubin test[62]. In most cases of our study, the MCMC chain converged after a burn-in of 1000 model cycles. For final inference, the parameters were sampled randomly ($n = 1000$) from the MCMC chains following an initial burn-in of 2000 cycles.

$$L = \frac{1}{\sqrt{2\pi}\sigma}\exp\left\{\frac{\sum(y - y_{model})^2}{2\sigma^2}\right\} \qquad (18)$$

where $L$ is the likelihood function; $y$ is the experimentally measured data (i.e., amount and $\delta^{13}C$ of butane pool); $y_{model}$ is the estimated value from model (i.e., first-order reaction model and logistic model); $\sum(y - y_{model})^2$ is the sum of squared error; $\sigma^2$ is the variance of error term. In the first-order reaction model, $\sigma$ was set arbitrarily to 5% of average $y$ (concentration of butane pool); its impact on model results were further assessed by sensitivity analysis (Supplementary Fig. 9). In logistic model, $\sigma$ was set conservatively at 1‰ to model the uncertainty in the measured $\delta^{13}C$ data.

## Sensitivity analysis

To assess how analytical choices affect the estimation of back flux extent ($\mu$), sensitivity analyses of isotope mass balance model were performed against variations in model parameters: (1) $\alpha_+$ (default value: 0.9975) was increased or decreased by 0.0002, to reflect its 95% confidence limits (Supplementary Fig. 5); (2) $\alpha_-$ (default value: 1) was varied from 0.90 to 1 (0.90, 0.95, and 1), with the assumption that carbon isotope fractionation associated with biological butane formation from $CO_2$ could be at the same order of magnitude as methanogenesis[63]; (3) $\sigma$, the error term in the likelihood function of first-order reaction model (Eq. 18), was increased from 5% (default) to 10%, or decreased to 2.5% of average $y$, to test the impact of data noise on model results; (4) initial concentration of total inorganic carbon

$([CO_2]_0)$ was varied by 10% of target values to account for the technical errors associated with substrate amendment using syringes.

## Dependence of AOB back flux extent on the availability of CO₂, butane and reducing equivalents

According to the flux force theorem, the back-to-forward flux ratio (f₋/f₊) is exponentially proportional to actual free energy of the AOB pathway ($\Delta G$; Eq. 19). Replacing forward flux (f₊) with net rate (f_net) yields the back flux extent (f₋/f_net) as a monotonic function of $\Delta G$ (Eq. 20). Equations 19 and 20 express that back flux of AOB relative to the forward flux or net rate is constrained not only by product to substrate quotient ($\{CO_2\}^4/\{C_4H_{10}\}$), but also the availability of reducing equivalents ($\{XH_2\}/\{X\}$). Higher product to substrate quotient leads to higher AOB back fluxes, whereas depletion of reducing equivalents declines its magnitude.

$$\frac{f_-}{f_+} = e^{\Delta G/RT} = \left(\frac{\{CO_2\}^4}{\{C_4H_{10}\}}\right) \times \left(\frac{\{XH_2\}^{13}}{\{X\}^{13}}\right) \times e^{\Delta G^{0'}/RT} \quad (19)$$

$$\frac{f_-}{f_{net}} = \frac{1}{e^{-\Delta G/RT} - 1} = \frac{1}{\left(\frac{\{C_4H_{10}\}}{\{CO_2\}^4}\right) \times \left(\frac{\{X\}^{13}}{\{XH_2\}^{13}}\right) \times e^{-\Delta G^{0'}/RT} - 1} \quad (20)$$

## Bioinformatics analyses

**Comparative genomics.** For comparative genomics, genomes of experimentally validated multicarbon alkane-oxidizing archaea[37] (ANKA; *n* = 9; Supplementary Data 1) and ANME[64] (*n* = 12; Data Table S1) were downloaded from NCBI Genome (https://www.ncbi.nlm.nih.gov/) and/or NODE (https://www.biosino.org/bmdc/) database. The genome completeness and contamination were assessed by CheckM 1.2.2 based on 188 Euryarchaeota marker genes. The protein-coding genes from each genome were predicted using prodigal v2.6.3 with -p meta option. All proteins were subsequently pooled and clustered into protein families following anvi'o pangenomic analysis pipeline[65]. In brief, pairwise similarity between protein sequences was calculated by running all-against-all blastp search with diamond (v0.9.24.125) --sensitive mode. The weak matches between sequences were eliminated with *minbit* heuristic of 0.5[66]. The resulting sequence similarity network was analyzed to cluster proteins into families, using the Markov cluster[67] program with an inflation parameter (-I) of 2.0. Protein families that are enriched in ANKA genomes over ANME genomes were screened for using scipy.stats.*fisher_exact()* function in Python v3.9. *P* values of Fisher Exact test were adjusted for multiple comparisons with scipy.stats.*false_discovery_control()* function in Python v3.9. *Ca.* S. butanivorans sequences associated with AOA-characteristic protein families were annotated with PFAM[68], KEGG[69] and COG[70] database. Cellular localization of the protein was predicted using Psortb v3.0[71]. The same procedure was applied to compare the ACR-encoding vs. MCR-encoding genomes within the class Syntropharchaeia. The representative Syntropharchaeia genomes (*n* = 61) were chosen based on GTDB taxonomy r220 and downloaded from NCBI database (Supplementary Data 2).

**Phylogenetic analysis of FrhB/FdhB in *Ca.* S. butanivorans.** Homologs of the beta subunit of coenzyme F420 hydrogenase (FrhB)/dehydrogenase (FdhB) encoded in the genome of *Ca.* S. butanivorans were retrieved using *hmmsearch* (HMMER 3.2.1; http://hmmer.org/), with the *--cut_ga* option. The hidden markov model (PF04432.hmm) for the *hmmsearch* was download from InterPro[72]/PFAM database[68], which targets the C-terminus of the FrhB/FdhB protein family. The yielded homologs in *Ca.* S. butanivorans were added to a reference phylogeny of FrhB/FdhB[64], which proposed 15 monophyletic clades of FrhB/FdhB from anaerobic methanotrophic (ANME) and

methanogenic archaea. For such phylogenetic placement analysis, *Ca.* S. butanivorans sequences were firstly added to the reference alignment using mafft (v7.407)[73], with the *--add* option. The resulting alignment was used to reconstruct maximum-likelihood (ML) tree with RAxML[74] version 7.2.6, using following parameters: raxmlHPC-PTHREADS-AVX2 -f a -m PROTGAMMALG -# autoMRE -p 12345 -T 24 -× 12345. The ML tree (Supplementary Fig. 10) was visualized and decorated with iTOL[75]. The neighboring genes of FrhB/FdhB homologs in *Ca.* S. butanivorans were plotted using the 'gggenes' package of R (version 4.1.0; https://CRAN.R-project.org/package=gggenes).

**Environmental distribution of ACR-encoding archaea.** The presence and relative abundance of ACR-encoding archaea across habitats were retrieved from the sandpiper database[76]. Genomes of ACR-encoding microorganisms were acquired from up-to-date overviews[37], and their taxonomy was extracted from the GTDB database[77] version r220 (Supplementary Data 5). The genus classification of GTDB taxonomy was then queried against the sandpiper database to yield metagenomes where the target genus has a relative abundance above 0.001%. The description and geographic location (latitude and longitude) for each metagenomic dataset were downloaded from the same database (Supplementary Data 8). The global distribution of ACR-encoding archaea was visualized in a world map using the *ggplot* package of R version 3.6.3.

The environmental distribution of AcrA sequences was explored using metagenomic data available in the JGI IMG/M database (May 2024 release)[78]. For such analysis, assembled genes encoding either the C-terminal (PF02249) or the N-terminal domains (PF02745) of MCR alpha subunit were retrieved from each metagenomic project. The yielded protein sequences (*n* = 102,813) were then placed to a reference AcrA tree[37] using RAxML evolutionary placement algorithm (EPA) with the following parameters: raxmlHPC-PTHREADS-AVX2 -T 60 -f v --epa-accumulated-threshold=0.95 -m PROTGAMMALG. To create the alignment for EPA, the sequences were added to the reference AcrA alignment using mafft (v7.407)[73], with the *--addfragments* option. Sequences that are placed to divergent AcrA branches with accumulated likelihood weight ratio over 0.95 were regarded as AcrA. The geographic location and type of environments of the sequences were retrieved or inferred from available metadata of the metagenomic project (Supplementary Data 9).

## Reporting summary

Further information on research design is available in the Nature Portfolio Reporting Summary linked to this article.

# Data availability

The bioinformatics, thermodynamic modelling, and isotope data generated in this study are provided in the Supplementary Information. The sequencing data are available in the NCBI database under the accession code BioProject ID PRJNA1134092. Source data are provided with this paper. Supplementary Data have been deposited in Figshare and are available at https://doi.org/10.6084/m9.figshare.25499440. Source data are provided with this paper.

# Code availability

The codes used for isotope (IsotopeModel) and metabolic (MetabolicModel) modelling were deposited and are publicly available at Github (https://github.com/SongCanChen11/BackFluxDuringAOB) and Zenodo (https://zenodo.org/records/13895334).

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

## Acknowledgements

We acknowledge J. Voigt for assistance with cultivation, sampling, and chemical analyses. Britta Poulsen and Susanne Nielsen are acknowledged for DNA extraction and amplicon sequencing. The experimental work was conducted at the former Department for Isotope Biogeochemistry, Helmholtz Centre for Environmental Research – UFZ, Leipzig, Germany. This study was funded by the Novo Nordisk Foundation through a Young Investigator Award (grant NNF22OC0071609 to F.M.), and by the Helmholtz Association of German Research Centers (S.C.C., N.M., J.J., S.K., O.J.L., H.H.R., F.M.). Additional financial support was provided by the Helmholtz Association through Grant ERC-RA-0020 (to F.M.). Song-Can Chen was supported by the Marie Skłodowska-Curie Actions 2021 program, European Commission, Belgium (Grant: 101059607 to S.C.C.). Sheng Chen was supported by the National Natural Science Foundation of China (NSFC) grant 12471341 (to S.C.) and the Foundation of Beijing Normal University grant 28704-312200502503 (to S.C.). We acknowledge the Centre for Chemical Microscopy (ProVIS) and the Laboratories for Stable Isotopes (LSI) at the Helmholtz Centre for Environmental Research for using their analytical facilities. ProVIS is supported by European Regional Development Funds (EFRE – Europe funds Saxony).

## Author contributions

S.C.C., H.-H.R., and F.M. designed research. S.C.C., J.J., and F.M. performed cultivation and stable isotope labeling experiments. S.C.C. and S.K. performed GC-IRMS analyses. N.M. performed probe design and testing, CARD-FISH, fluorescence microscopy and cell counting. M.B.L performed amplicon sequencing and bioinformatics analyses. N.M., and F.M. performed physiology and net back flux assays. S.C.C., A.G., O.J.L., and H.-H.R. developed a conceptual model for data analysis. S.C.C. and S.C. developed the mathematical model for data interpretation. S.C.C. performed bioinformatics analyses. S.C.C. and F.M. wrote the manuscript with contributions from all co-authors.

## Competing interests

The authors declare no competing interests.
