## [Transparent Peer Review file · Nature Communications]

Back flux during anaerobic oxidation of butane support archaea-mediated alkanogenesis

Corresponding Author: Dr Florin Musat

This manuscript has been previously reviewed at another journal. This document only contains reviewer comments, rebuttal and decision letters for versions considered at Nature Communications.

Version 0:

Reviewer comments:

Reviewer #1

(Remarks to the Author)

The authors have generally addressed the main concerns raised. I only have a few minor comments to the current manuscript.

1. L64-65: the authors showed that the relative abundance of *Ca. Desulfofervidus* is 30% based on amplicon sequencing. For the same sample, what is the abundance of the SRB based on CARD-FISH results?
2. L250: "A" should be "As".
3. L275: in the back flux experiments, butane was introduced to the serum bottles with three different levels. However, the initial levels of butane among different groups are similar. Why? In addition, if different levels of butane were introduced to the serum bottles, the microbial community might change in those different groups at the end of the long-term experiments, particularly for *Ca. S. butanivorans* and *Ca. S. caldarius*. However, the information is missing.
4. L299: the sampling information should be provided for the samples used for CARD-FISH.

Reviewer #2

(Remarks to the Author)

thank you for addressing my concerns about the community members.

One thing arising out of the revision, the colors used to describe the FISH images do need to be better defined. I would use magenta instead of red-purple description, it differentiates it from those called purple cells. Also, use cyan for the blue-green description, again it better differentiates it from green cells.

Reviewer #3

(Remarks to the Author)

This manuscript has been revised substantially to address the reviewer's feedback. The microbial compositions via 16S rRNA amplicon profiling as well as CARD-FISH has been very informative in understanding the putative members participating in trace butane formation. *Ca. S. butanivorans* was the most dominant community member mediating AOB. The inclusion of the replicates used was also very helpful in assessing the viability of the study. These additional inputs provide compelling evidence of trace butane formation, and therefore the reversibility of butane oxidation coupled to sulfate reduction. The formation of longer chain alkanes via an ACR-dependent pathway has not been demonstrated before and thus provides a novel finding. Much like methanogens and ANME, the potential of alkanogenic microbes is certainly interesting and should be further investigated.

I have no issue with this manuscript and support its acceptance.

A few minor comments:

Could you supply the 16S rRNA amplicon results for supplementary figure 1 as a table?

Extended figure 6a: screening of IMG/M for ACR variants is greatly appreciated. Were there ACR sequences identified that were distantly related to those found in your reference tree?

The Extended Figures 6c and 6d plots are squished, adjusting these for better visualization and interpretation.

As mentioned in the discussion, could the authors comment on whether backfluxes for even longer-chain alkanes beyond butane are feasible?

Response letter for manuscript *Back flux during anaerobic oxidation of butane support archaea-mediated alkanogenesis*, by Chen and coauthors, for consideration at Nature Communications

Author reply. We acknowledge all reviewers for their constructive comments, and for supporting publication of our manuscript.

Remaining reviewer comments:

Reviewer #1:

The authors have generally addressed the main concerns raised. I only have a few minor comments to the current manuscript.

1. L64-65: the authors showed that the relative abundance of *Ca. Desulfofervidus* is 30% based on amplicon sequencing. For the same sample, what is the abundance of the SRB based on CARD-FISH results?

Author reply. With our focus on distinguishing between *Ca. S. butanivorans* and *Ca. S. caldarius*, and determining the abundance of these archaeal clades (as the bearers of the butane oxidation pathway), we have not investigated the abundance of *Ca. Desulfofervidus* by CARD-FISH. The value of 30% we have obtained here based on amplicon sequencing is very similar to the metagenome reads-based abundance obtained formerly (29.6%; Laso-Perez et al., Nature 2016).

2. L250: "A" should be "As".

Author reply. Corrected.

3. L275: in the back flux experiments, butane was introduced to the serum bottles with three different levels. However, the initial levels of butane among different groups are similar. Why? In addition, if different levels of butane were introduced to the serum bottles, the microbial community might change in those different groups at the end of the long-term experiments, particularly for *Ca. S. butanivorans* and *Ca. S. caldarius*. However, the information is missing.

Author reply. This was a misleading phrasing on our side. In an attempt to compress the method description, we have listed here all different concentrations of butane used across all our experiments. All our main back flux assays were provided with the same volume of butane (9 ml). We used the different volumes of butane listed when we tested the impact of the substrate concentration (butane) on the back flux extent (Figure 5). The description was revised for clarity.

We appreciate that different substrate levels do not affect the microbial communities of slow-growing hydrocarbon-degrading cultures like this one. Instead, kinetics are affected by mass transfer limitations. These are cultures with one single growth substrate: when that is limiting, the entire culture metabolism will be limited, including that of microorganisms dependent on the primary butane degrader.

4. L299: the sampling information should be provided for the samples used for CARD-FISH.

Author reply. Samples were collected from 3 independent cultures ahead of the back flux assay. The description was updated to include this information.

Reviewer #2:

thank you for addressing my concerns about the community members.

One thing arising out of the revision, the colors used to describe the FISH images do need to be better defined. I would use magenta instead of red-purple description, it differentiates it from those called purple cells. Also, use cyan for the blue-green description, again it better differentiates it from green cells.

Author reply. We have updated the description in Figure 1 legend according to the reviewer's suggestion.

Reviewer #3:

This manuscript has been revised substantially to address the reviewer's feedback. The microbial compositions via 16S rRNA amplicon profiling as well as CARD-FISH has been very informative in understanding the putative members participating in trace butane formation. *Ca. S. butanivorans* was the most dominant community member mediating AOB. The inclusion of the replicates used was also very helpful in assessing the viability of the study. These additional inputs provide compelling evidence of trace butane formation, and therefore the reversibility of butane oxidation coupled to sulfate reduction. The formation of longer chain alkanes via an ACR-dependent pathway has not been demonstrated before and thus provides a novel finding. Much like methanogens and ANME, the potential of alkanogenic microbes is certainly interesting and should be further investigated.

I have no issue with this manuscript and support its acceptance.

A few minor comments:

Could you supply the 16S rRNA amplicon results for supplementary figure 1 as a table?

Author reply. We now supply the amplicon sequencing results as a Source Data table.

Extended figure 6a: screening of IMG/M for ACR variants is greatly appreciated. Were there ACR sequences identified that were distantly related to those found in your reference tree?

Author reply. All reported AcrA variants could be placed to the reference AcrA clade with high confidence. Nevertheless, our screening procedure indeed placed some IMG/M homologs between the McrA and AcrA clade, which potentially represent McrAs and/or AcrAs that are distantly related to the reference clades. However, due to the fragmented nature of the assembled contigs and the uncertainty of their evolutionary placement, we can't reliably assign these homologs as divergent AcrA.

The Extended Figures 6c and 6d plots are squished, adjusting these for better visualization and interpretation. As mentioned in the discussion, could the authors comment on whether backfluxes for even longer-chain alkanes beyond butane are feasible?

Author reply. The plots in ED Figure 6 were adjusted. Since Nature Communications does not allow extended data, this figure was now moved into the main text as Figure 6.

We believe that back fluxes of even longer chain alkanes are mechanistically feasible, however it will be challenging to probe this experimentally due to label dilution along with chain elongation. We would refrain from including such comments in the present manuscript, since we find them highly speculative and not supported by our data.